# An Improved Approach towards Multi-Agent Pursuit–Evasion Game Decision-Making Using Deep Reinforcement Learning

**DOI:** 10.3390/e23111433

**Published:** 2021-10-29

**Authors:** Kaifang Wan, Dingwei Wu, Yiwei Zhai, Bo Li, Xiaoguang Gao, Zijian Hu

**Affiliations:** School of Electronics and Information, Northwestern Polytechnical University, Xi’an 710072, China; wudingwei@mail.nwpu.edu.cn (D.W.); zyw@mail.nwpu.edu.cn (Y.Z.); libo803@nwpu.edu.cn (B.L.); cxg2012@nwpu.edu.cn (X.G.); huzijian@mail.nwpu.edu.cn (Z.H.)

**Keywords:** pursuit–evasion, multi-agent, deep reinforcement learning, decision-making, adversarial learning, MADDPG

## Abstract

A pursuit–evasion game is a classical maneuver confrontation problem in the multi-agent systems (MASs) domain. An online decision technique based on deep reinforcement learning (DRL) was developed in this paper to address the problem of environment sensing and decision-making in pursuit–evasion games. A control-oriented framework developed from the DRL-based multi-agent deep deterministic policy gradient (MADDPG) algorithm was built to implement multi-agent cooperative decision-making to overcome the limitation of the tedious state variables required for the traditionally complicated modeling process. To address the effects of errors between a model and a real scenario, this paper introduces adversarial disturbances. It also proposes a novel adversarial attack trick and adversarial learning MADDPG (A2-MADDPG) algorithm. By introducing an adversarial attack trick for the agents themselves, uncertainties of the real world are modeled, thereby optimizing robust training. During the training process, adversarial learning was incorporated into our algorithm to preprocess the actions of multiple agents, which enabled them to properly respond to uncertain dynamic changes in MASs. Experimental results verified that the proposed approach provides superior performance and effectiveness for pursuers and evaders, and both can learn the corresponding confrontational strategy during training.

## 1. Introduction

With the development of the three generations of artificial intelligence [1], the technology of multi-agent systems (MASs) has been widely used in many areas of society, such as multi-agent motion planning, complex IT systems, computer communication technology, and so on [2,3,4,5]. Pursuit–evasion games have been widely investigated in MASs during recent years. They have been extended to various fields, to include maneuvering target tracking, surveillance early warning, anti-intrusion protection, and intelligent transportation [6,7]. The goal of these studies is to provide good strategies for pursuers and evaders. For pursuers, their goal is to round up the evaders as much as possible through cooperative decision-making. For evaders, they need to choose the best strategy based on the actions of pursuers to design an escape path to prevent being captured [8].

To address this problem, a series of research activities on agent-based pursuit–evasion games has been carried out in the differential gaming field. Isaacs [9] proposed a one-to-one robot hunting problem where partial differential equations describing the pursuer and the evader were created and solved analytically. Furthermore, a generalized maximum–minimum solving method of the Hamilton Jacobi equation for pursuit–evasion games was provided by Krasovskii [10]. Because in complex control problems, directly solving differential equations is very complicated and consumes many computing resources, researchers proposed some intelligent optimization algorithms that provide new ideas for solving the differential equation problems associated with pursuit–evasion games. Chen et al. [11] simulated fish foraging behavior and proposed a cooperative pursuit strategy that studied pursuit and evasion when trackers have a constrained turning rate. Wang et al. [12] introduced an alliance generation algorithm that generates a synergistic strategy based on the emotional factors of multirobot systems. This ensured that a team’s agents worked towards a common goal. However, there are many constraints and state variables involved in the complicated control process governing these issues, which make the solution intricate especially in a complex and dynamic scenario with multi-agent confrontation. Therefore, more intelligent algorithms are needed to effectively solve the problem of pursuit–evasion games.

By combining deep learning’s ability to perceive highly dimensional data [13] and reinforcement learning’s decision-making ability [14], deep reinforcement learning (DRL) provides a new optimization scheme for intelligent decision-making or control. Because techniques based on deep reinforcement learning do not require the establishment of a differential game model and agents can learn the optimal confrontation strategy only through interaction with the environment [15], some scholars introduced deep reinforcement learning in pursuit–evasion games and acquired the Nash equilibrium of the problem. Xu et al. [16] established a multi-agent reinforcement learning model for UAV pursuit–evasion in which relative motion state equations were employed. As a result, the pursuit–evasion issue was converted into a zero-sum game addressed through minimax-Q learning. In predatory games, Park et al. [17] set up a co-evolution framework for predator and prey to allow multiple agents to learn good policies by deep reinforcement learning. Gu et al. [18] presented an attention-based fault-tolerant model, which could also be applied to pursuit–evasion games, and the key idea was to utilize the multihead attention mechanism to select the correct and useful information for estimating the critics. To solve the complicated training problems caused by discrete action sets introduced by deep Q networks [19], Liu et al. [20] transformed a space rendezvous optimization problem between a space vehicle and noncooperative target into a pursuit–evasion differential game. They introduced a branching architecture with a group of parallel neural networks and shared decision modules. To overcome the unstable recognition ability of pursuers, Qadir et al. [21] proposed a novel approach for self-organizing feature maps and deep reinforcement learning based on the agent group role membership function model. Experiments verified the effectiveness of this method for facilitating the capture of evaders by mobile agents. Singh et al. [22] built on the actor–critic model-free multi-agent deep deterministic policy gradient algorithm to operate over the continuous spaces of pursuit–evasion games. In their approach, the evader’s strategy is not learned. It is based on Voronoi regions that pursuers try to minimize and evaders try to maximize.

Although they represent progress, previous studies on DRL-based pursuit–evasion games are still in their early stages. In these studies, pursuing platforms are assumed to be equipped with error-free identification and measurement systems that allow them to acquire precise information about the position, velocity, and other characteristics of evaders and cooperators [6,23]. However, sensors and other equipment configured in an unmanned system encounter positioning, sensing, and actuator error in reality [24,25]. These errors cause the environment to become uncertain, thereby affecting the strategies of the pursuers and evaders and making their performance worse. Therefore, this research is about designing a robust algorithm for MASs to effectively mitigate these errors and that would be significant for application research in real-world multi-agent decision-making.

This paper introduces a novel multi-agent algorithm to address the decision-making problem of pursuit–evasion games. The algorithm can solve pursuit–evasion games in complex virtual and real environments, where there are static or moving obstacles and pursuers and evaders need to avoid them while making decisions. Specifically, we make the following contributions in this paper:

(1) We develop an actor–critic-based motion control framework based on the multi-agent deep deterministic policy gradient (MADDPG) [26], which can take the state and behavior of other partners into account and is used to provide collaborative decision-making capabilities for each agent in the MAS;

(2) We propose an advanced algorithm called A2-MADDPG, which uses two skills to make the training strategy robust. The first is adversarial attack tricks for agents. It proposes to sample the status after stochastic Gaussian noise is applied, and this approach can train a robust agent to cope with measurement errors in the real world. The second is the optimized adversarial learning technique [27]. It is introduced to improve agent stability and to assist in adapting to noise produced by interactions between multiple agents;

(3) We verified the effectiveness and robustness of the algorithm in simulation experiments. We compared the performance of the proposed method with two common and advanced algorithms, namely the MADDPG and the independent multiagent deep deterministic policy gradient (IMDDPG), where the IMDDPG is a natural extension of the DDPG [28] in the field of multi-agents. Through a series of experiments, we show that the proposed method presents excellent performance for both pursuers and evaders compared with the MADDPG and IMDDPG in the case of the same hyperparameter settings and simulation environment parameter settings, and it can help them both develop robust motion strategies.

The rest of the paper is structured as follows: Section 2 provides background information about multi-agent pursuit–evasion games and describes related theoretical approaches. Section 3 introduces a framework for collaborative pursuit missions and an improved A2-MADDPG algorithm where an adversarial attack trick and an adversarial learning-based optimization method are combined with the MADDPG. Section 4 verifies the robustness and high performance of the algorithm through simulation experiments. Section 5 provides a conclusion and envisages future work.

## 2. Background

In this section, the kinematic and observation model of agents executing a pursuit–evasion task is presented. In addition, the essential theoretical background of the DRL-based MADDPG algorithm and adversarial learning is introduced.

### 2.1. Problem Definition

The multi-agent pursuit–evasion game problem can be described as follows: there are pursuers (red agents) and evaders (blue agents), as shown in Figure 1. Both agent types have different tasks based on their maneuverability. Each agent can perceive the relative position of the threat zone (gray circle) using radar and sensors. The velocity and position of each agent are provided by its navigation equipment, and they share information by transmitting through a signaling connection. Pursuers are equipped with an attack or shielding interference device (the red circle represents the attack range of the pursuit), and their mission is considered successful when they suppress an evader by approaching it. Evaders must stay away from pursuers. Neither pursuers nor evaders can exceed their boundaries.

#### 2.1.1. Comparisons of Operators

A general decision-making program for pursuit–evasion gaming is primarily used to determine the communication and cooperation between platforms and achieve target pursuit. This is performed without fully considering the maneuvering characteristics of the platforms. Both agents in this paper are mobile UAVs flying at a fixed altitude with nonholonomic constraints [29], as portrayed in Figure 2. The status update of each UAV can be described as:(1)$$\left\{\begin{array}{c} p^{t}=\left[\begin{array}{c} x^{t-1}+v_{x}^{t}\cdot \Delta t\\ y^{t-1}+v_{y}^{t}\cdot \Delta t \end{array}\right]\\ v^{t}=\left[\begin{array}{c} v_{x}^{t-1}+a_{x}^{t}\cdot \Delta t\\ v_{y}^{t-1}+a_{y}^{t}\cdot \Delta t \end{array}\right]\\ \varphi^{t}=a\tan\left(\frac{v_{y}^{t}}{v_{x}^{t}}\right) \end{array}\right.$$
where $p^{t}$, $v^{t}$, and $\varphi^{t}$ denote the position, velocity, and yaw angle parameters. The superscript *t* represents time *t*; Δt is the time interval; *a* is UAV’s acceleration. Considering power systems and mechanical limitations, the maximal velocity and acceleration are assumed to be $v_{max}$ and $a_{max}$, which are introduced in the following simulation.

#### 2.1.2. Observation Model

The observation model of the agent was presented to provide the agent with the ability to sense the environment [30]. In this multi-agent pursuit–evasion task, $\left(x_{a}^{i},y_{a}^{i}\right)$ is the position of each agent in the pursuit formation, and both $\left(x_{e}^{j},y_{e}^{j}\right)$ and $\left(x_{o}^{k},y_{o}^{k}\right)$ represent the position of the center point of the evader and the threatened area, respectively. The number of pursuers, evaders, and obstacles in the environment is defined as num_P, num_E, and num_o, respectively. Since pursuers and evaders need to consider avoiding obstacles to prevent being hit when making decisions, these obstacles make it more difficult to solve the problem of pursuit–evasion. The formation of all pursuers is denoted as *A*. An agent *i* on the pursuers’ team can use radar detection and communication transmission to obtain its own local observations from the environment as follows:(2)$$\begin{array}{c} \left\{\begin{array}{c} O_{s}^{i}=\left[v_{s}^{i},p_{s}^{i}\right]=\left[v_{sx}^{i},v_{sy}^{i},x_{s}^{i},y_{s}^{i}\right]\\ O_{c}^{i}={\left[p_{s}^{l}\right]}_{l=1,\ldots ,i-1,i+1,num\_P}={\left[x_{c}^{l},y_{c}^{l}\right]}_{l=1,\ldots ,i-1,i+1,num\_P}\\ O_{e}^{i}={\left[p_{s}^{j}\right]}_{j=1,\ldots ,num\_E}={\left[x_{e}^{j},y_{e}^{j}\right]}_{j=1,\ldots ,num\_E}\\ O_{o}^{i}={\left[p_{o}^{k}\right]}_{k=1,\ldots ,num\_o}={\left[x_{o}^{k},y_{o}^{k}\right]}_{j=k,\ldots ,num\_o} \end{array}\right. \end{array}$$

Here, $v_{s}x^{i}$, $v_{s}y^{i}$, $x_{s}^{i}$, and $y_{s}^{i}$ represent the self-observed velocity and position of the agent on the x and y axis. $O_{c}^{i}$ indicates the observed location of other pursuers in the formation, and *l* is the sequence number of other pursuers on the team. $O_{e}^{i}$ denotes the observed location of an evader, and *j* represents its sequence number. $O_{o}^{i}$ represents information observed about an obstacle, and *k* is the obstacle number. Considering a real mission scenario, a set of range sensors is employed to help the unmanned system detect possible threats from obstacles ahead of it in the range. As shown in Figure 3, the 90${}^{\circ }$ angle containing the blue arc within the sensor range is the agent’s threat detection area. An agent’s observations about an obstacle are divided into five parts:(3)$$O_{o}^{i}=\left[d_{1},d_{2},d_{3},d_{4},d_{5}\right]$$
where $d_{1-5}$ denotes the five sensor indications. We set $d_{1-5}=L$ when a threat is not detected. Based on the comprehensive observation information above, an agent can perceive and assess the situation.

### 2.2. Theoretical Context

Deep reinforcement learning is a representative intelligent machine learning algorithm, and adversarial learning can increase the stability and robustness of the model trained by reinforcement learning [31]. They provide new research ideas for multi-agent pursuit–evasion decision-making. In this section, adversarial learning, the DRL-based DDPG algorithm, and the MADDPG algorithm are introduced.

#### 2.2.1. Adversarial Learning

Adversarial learning is a technique of defending against adversarial samples [32]. This approach attempts to improve the accuracy of neural network models by training adversarial samples and normal samples together and reducing the interference of the adversarial samples. The robustness and generalization ability of the resulting network are improved. Adversarial training can be expressed as follows:(4)$$\min_{\theta }\;\max_{D\left(x,x^{\prime }\right)<\eta }\;L_{adv}\left(x^{\prime },y;\theta \right)$$
where *x*, $x^{\prime }$ denote the original sample and adversarial sample, respectively, *y* is the label value, and θ is the weight of the networks. $D\left(x,x^{\prime }\right)$ represents the distance measurement between the original sample and the adversarial sample, and $L_{adv}\left(x^{\prime },y;\theta \right)$ represents the adversarial loss function. In the min–max form, the internal maximization optimization problem is to find the optimal adversarial sample, and the external minimization optimization problem is to minimize the loss function. The learning process of confrontation training is depicted in Figure 4.

The fast gradient sign method (FGSM) efficiently generates adversarial samples [33]. The FGSM uses a model’s objective loss function to determine the input vector needed to calculate its counter disturbance, which it adds to the corresponding input. This generates counter samples that correspond to the original samples. Suppose that in a classification problem, the output label of the model is $class=\left\{0,1\right\}$. After adversarial training, the model will have higher prediction confidence, that is the model will output the correct sample label even if a small adversarial disturbance is added to the sample. This process can be defined as:(5)$$x^{*}=x+\eta_{adv}$$
(6)$$class\left(x^{*}\right)=class\left(x\right)$$
where $\eta_{adv}$ represents the sample perturbation added and $x^{*}$ represents adversarial samples after adding perturbation. Each time the model is trained, the FGSM performs an optimization along the gradient direction of the counter loss function $L_{adv}\left(x^{\prime },y;\theta \right)$, and counter samples are obtained. The generation process of sample disturbance $\eta_{adv}$ can be expressed as:(7)$$\eta_{adv}=-\varepsilon \frac{g}{{∥g∥}_{2}}$$
(8)$$g=\nabla_{x}L_{adv}\left(x^{\prime },y;\theta \right)$$
where ε denotes the magnitude of disturbance and *g* is the inverse gradient of the input vector. Moreover, the FGSM-based target loss function can be defined as:(9)$$J\left(x,y;\theta \right)=cJ_{adv}\left(x^{*},y;\theta \right)+\left(1-c\right)J_{raw}\left(x,y;\theta \right)$$
where *c* is an equilibrium coefficient that is used to balance both the original and attack samples. As a result, the adversarial examples used in the adversarial learning method can improve the generalization ability of a model by adding a regular term to the loss function. The goal of adversarial training is to minimize the loss function in the worst case.

#### 2.2.2. DRL-Based DDPG Algorithm

During the deep reinforcement learning process, an agent completes its interaction with the environment by perceiving the environment and taking appropriate actions. It performs adaptive iterative optimization according to a reward signal, as shown in Figure 5. An effective approach to describe the DRL-based training process is the Markov decision process (MDP) [34], which is represented by a five-tuple $\langle S,A,R,P,\gamma \rangle$. At each time step, an agent interacts with the environment and makes observations, which comprise the agent’s state s∈S. Agents then perform the action a∈A to obtain reward *R* from *s* to a new state $s^{\prime }$. *P* denotes the environmental model, and it represents the probability distribution of transitioning to a new state. γ is a discount factor used to balance the impact of instantaneous returns and long-term returns on cumulative rewards.

The deep deterministic policy gradient is an algorithm that combines policy-based actor neural networks with value-based critic neural networks that can be employed for continuous control [28]. The actor online network μ reacts according to the agent’s current observation state $s_{t}$ and generates a reasonable action $a_{t}=\mu \left(s_{t}\right)$. The critic online network *Q* is responsible for evaluating the current action and outputting the action value function $Q(s_{t},a_{t};\theta^{Q})$. $\theta^{\mu }$ and $\theta^{Q}$ denote the corresponding parameters of an actor online network and a critic online network. In addition, actor target networks $\mu^{\prime }$ and critic target networks $Q^{\prime }$ are constructed for future updates.

After each decision, a training sample $\left[s_{t},a_{t},r_{t},s_{t}^{\prime }\right]$ for time *t* is collected in the experience buffer *M*, which is applied to iteratively improve the agent’s strategies, that is in the update optimization phase, a stochastic mini-batch of *N* arrays of samples of the previous format is extracted for every training time. The critic online network is updated according to the TD-error, which is defined as:(10)$$L\left(\theta^{Q}\right)=E_{s,a,r,\;s^{\prime }}\left[{\left(y-Q^{\mu }\left(s,a;\theta^{Q}\right)\right)}^{2}\right]$$
(11)$$y=r+{\left.\gamma Q^{\mu^{\prime }}\left(s^{\prime },a^{\prime };\theta^{Q^{\prime }}\right)\right|}_{a^{\prime }=\mu^{\prime }\left(s\right)}$$

Here, $L\left(\theta^{Q}\right)$ is the loss function of critic networks, *y* is the target value Q-target, and *i* represents the sequence of extracted samples. Additionally, the actor online network would be trained by minimizing the following policy gradient:(12)$$\nabla_{\theta^{\mu }}J=E_{s,a∼M}\left[\nabla_{\theta^{\mu }}\mu \left(a\left|s\right.\right)\nabla_{a}Q^{\mu }\left(s,a;\theta^{Q}\right)\left|a=\mu \left(s\right)\right.\right]$$

At regular intervals, the soft update approach and update factor τ are used to copy the network parameters to the target network, which can be expressed as:(13)$$\left\{\begin{array}{c} \theta^{Q^{\prime }}=\tau \theta^{Q}+\left(1-\tau \right)\theta^{Q^{\prime }}\\ \theta^{\mu^{\prime }}=\tau \theta^{\mu }+\left(1-\tau \right)\theta^{\mu^{\prime }} \end{array}\right.$$

#### 2.2.3. Multi-Agent Deep Deterministic Policy Gradient Algorithm

The MADDPG algorithm is an effective DRL algorithm derived from the DDPG algorithm and can be used to address problems with multi-agent strategies. In the MADDPG, each agent has its own actor–critic framework [35]. For a multi-agent system, the observation set consisting of *n* agents is $x=\left(s_{1},s_{2},\ldots ,s_{n}\right)$, the action set is $a=\left(a_{1},a_{2},\ldots ,a_{n}\right)$, and the reward set is $r=\left(r_{1},r_{2},\ldots ,r_{n}\right)$. For each agent, its observations and actions are denoted as $s_{i}$ and $a_{i}=\mu_{i}\left(s_{i}|\theta_{i}^{\mu }\right)$ at a point in time. The agent’s actor online network outputs a policy according to the agent’s own observations, and its critic online network estimates a centralized action value function $Q_{i}^{\mu }\left(x,a_{1},a_{2},\ldots ,a_{n}\right)$. This function is based on the status and actions of all agents, as depicted in Figure 6.

During each interaction within the environment, an agent will store relevant experiences in the experience buffer. Unlike the DDPG, the *N* comprehensive learning samples $\left[x,a_{1},a_{2},\ldots ,a_{n},r_{1},r_{2},\ldots ,r_{n},x^{\prime }\right]$ in the MADDPG are drawn and spliced from the experience buffer of all agents each time one is trained. For agent *i*, the critic online network is updated according to:(14)$$L\left(\theta_{i}\right)=E_{x,a,r,x^{\prime }}\left[{\left(y_{i}-Q_{i}^{\mu }\left(x,a_{1},\ldots ,a_{n};\theta_{i}^{Q}\right)\right)}^{2}\right]$$
(15)$$y_{i}=r_{i}+\gamma Q_{i}^{\mu^{\prime }}\left(x^{\prime },a_{1}^{\prime },\ldots ,a_{n}^{\prime };\theta_{i}^{Q^{\prime }}\right)\left|{}_{a_{i}=\mu_{i}\left(s_{i}\right)}\right.$$

Meanwhile, by minimizing the policy gradient, its actor online network can be optimized. This is expressed as:(16)$$\nabla_{\theta_{i}^{\mu }}J=E_{x,a∼D}\left[\nabla_{\theta_{i}^{\mu }}\mu_{i}\left(a_{i}\left|s_{i}\right.\right)\nabla_{a_{i}}Q_{i}^{\mu }\left(x,a_{1},\ldots ,a_{n};\theta_{i}^{Q}\right)\left|{}_{a_{i}=\mu_{i}\left(s_{i}\right)}\right.\right]$$

The MADDPG algorithm also borrows the soft update technique described in Equation (Equation 13) from DDPG.

Although agents trained with the MADDPG can achieve good results in some simple environments, the multi-agent system is very sensitive to changes in the training environment, and the convergence strategies obtained by agents are likely to fall into a local optimum, that is when the strategies of other agents change, the agent cannot produce the optimal action strategy. In order to improve the robustness of the strategy, this paper combines the MADDPG and adversarial learning to propose the A2-MADDPG algorithm, which is introduced in Section 3.

## 3. Proposed Method

This section proposes an approach for realizing control for pursuers and evaders in a game that contains an uncertain environment. There are obstacles in the environment that both pursuers and evaders need to avoid, and when acquiring specific values in the state space, sensors and other devices encounter positioning, sensing, and actuator errors, resulting in inaccurate values, so the environment is uncertain for pursuers and evaders. An MADDPG-based control framework for multi-agent systems is presented, and it includes action, state, space, and specific reward functions. Furthermore, an improved approach called the A2-MADDPG is described. The A2-MADDPG incorporates an adversarial attack trick and adversarial learning into the MADDPG algorithm.

### 3.1. MADDPG-Based Framework

#### 3.1.1. Actor Space

When addressing DRL-based multi-agent decision-making, state and action spaces must be defined based on the MDP framework. To ensure that mission control is more similar to the real world, UAVs use dual-channel control, that is the force on a UAV is controlled directly. The effects of this force are then applied to the UAV’s movement attitude and flight velocity. The action output $A^{i}$ of a dual-channel thrust UAV thrust can be expressed as:(17)$$A^{i}=\left[F_{x}^{i},F_{y}^{i}\right]$$
where the superscript *i* represents the sequence number of the UAV in an MAS. $F_{x}^{i}$, $F_{y}^{i}$ represent the force on the x and y axes that the UAV received. Therefore, the acceleration can be given by:(18)$$a^{i}=\left[\begin{array}{c} a_{x}^{i}\\ a_{y}^{i} \end{array}\right]=\left[\begin{array}{c} F_{x}^{i}\\ F_{y}^{i} \end{array}\right]/m_{u}$$
where $m_{u}$ represents the mass of the UAV. The UAV’s attitude can then be adjusted when combined with Equation (Equation 1).

#### 3.1.2. State Space

The state space of a UAV provides useful information based on an agent’s observation model. This is used to help the agent sense its surroundings and make decisions. To help both sides during confrontation training, the state spaces of pursuers and evaders should be presented. As per Section 2.1.2, each pursuer’s state information is processed and integrated and includes its position relative to partners $s_{pc}^{i}={\left[x_{pc}^{l}-x_{ps}^{i},y_{pc}^{l}-y_{ps}^{i}\right]}_{l=1,\ldots ,i-1,i+1,num\_P}$, evader targets $s_{pe}^{i}={\left[x_{pe}-x_{ps}^{i},y_{pe}^{}-y_{ps}^{i}\right]}_{j=1,\ldots ,num\_E}$, and obstacles $s_{po}^{i}=\left[d_{1}^{i},\ldots ,d_{5}^{i}\right]$, as mentioned in Equation (Equation 3). The state space of pursuer *i* can be defined as:(19)$$S_{p}^{i}=\left[s_{ps}^{i},s_{pc}^{i},s_{pe}^{i},s_{po}^{i}\right]$$
where $s_{ps}^{i}=\left[v_{px}^{i},v_{py}^{i},x_{ps}^{i},y_{ps}^{i}\right]$ denotes its position and speed based on self-observed information that has not been processed. Similarly, the state space of an evader *j* can be defined as:(20)$$S_{e}^{j}=\left[s_{es}^{j},s_{ec}^{j},s_{ep}^{j},s_{eo}^{i}\right]$$
where $s_{es}^{j}=\left[v_{ex}^{j},v_{ey}^{j},x_{es}^{j},y_{es}^{j}\right]$ represents the position and speed of evader *j*, $s_{ec}^{j}$ represents its position relative to other evaders, $s_{ec}^{j}={\left[x_{ec}^{l}-x_{es}^{j},y_{ec}^{l}-y_{es}^{j}\right]}_{l=1,\ldots ,j-1,j+1,num\_E}$$s_{ep}^{j}={\left[x_{ep}^{i}-x_{es}^{j},y_{ep}^{i}-y_{es}^{j}\right]}_{i=1,\ldots ,num\_P}$ represents its position relative to pursuers, and $s_{eo}^{j}=\left[d_{1}^{j},\ldots ,d_{5}^{j}\right]$ represents sensed obstacles.

#### 3.1.3. Reward Function

In the traditional MADDPG algorithm, formation cooperation cannot be uniformly controlled since each agent has an independent actor and critic network. When a unit successfully hunts down a target, all agents belonging to the formation receive a positive reward regardless of whether the agent was in effective tracking range or played a positive role in the mission. This is contrary to the incentive policy of real pursuit–evasion scenarios.

To address this problem, a reward function based on the team strategy was presented. An agent could receive a positive reward only if it was within a certain distance $\zeta range_{att}$ of the target when the mission terminated. The reward is shaped by three basic elements: (1) distance $r_{distance}^{i}$: the Euclidean distance is used to judge whether the agent successfully pursued the evader; (2) maneuvering safety $r_{safe}^{i}$: the agent is punished if it has collided with obstacles or collaborators; and (3) mission criteria $r_{mission}^{i}$, $\zeta range_{att}$ are used to judge whether the agent completed the mission. These three subreward functions can be defined as:(21)$$r_{distance}^{i}=dis\left(p_{p}^{i}-p_{e}^{j}\right)$$
(22)$$r_{safe}^{i}=\left\{\begin{array}{c} 1,\;dis\left(p_{p}^{i}-p_{o}^{k}\right)>dis_{obstacle}\;or\;dis{\left(p_{p}^{i}-p_{p}^{n}\right)}_{n\neq i}>dis_{safe}\\ 0,\;else \end{array}\right.$$
(23)$$r_{safe}^{i}=\left\{\begin{array}{c} 1,\;dis\left(p_{p}^{i}-p_{e}^{j}\right)\leq \zeta range_{att}\\ 0,\;else \end{array}\right.$$
where *i* is the sequence number of the agent and function dis(a,b) is used to calculate the Euclidean distance of positions *a* and *b*. $dis_{obstacle}$ represents the radius of an obstruction, and $dis_{safe}$ represents the minimum safe distance between each pursuer. To summarize, the reward function for a pursuer *i* can be formulated as:(24)$$r_{p}^{i}=\beta_{1}r_{distance}^{i}+\beta_{2}r_{final}^{i}+\beta_{3}r_{safe}^{i}$$

Three relative gain factors $\beta_{1-3}$ are introduced, which represent the respective weights of the three rewards or punishments. Among them, $\beta_{1}$ is negative, and both $\beta_{2}$ and $\beta_{3}$ are positive. To train an autonomous evader, a specific reward function was built according to the distance among the pursuer, evaders, and obstacles, and it has weights that are the inverse of the pursuer’s reward function.

### 3.2. A2-MADDPG Algorithm

#### 3.2.1. Adversarial Attack Trick for the Agent

When perceiving the environment in a real scenario, an agent would encounter unavoidable errors due to the detection process, image recognition, signal processing, and satellite position-based parameter measurements. Improving model robustness is of great significance, especially in key intelligent control fields such as UAVs and robots, where tiny errors or noise could lead to immeasurable and undesirable consequences.

To train a robust agent to adapt to measurement errors and other noise in real environments, an adversarial attack trick for the agent itself is proposed. This approach aims to generate random noise in the agent’s status, thereby confusing its perception and assisting it in producing a strategy for abnormal conditions [36]. Algorithm 1 summarizes the adversarial attack process, in which inputs are constituted by the action $a_{i}=\mu_{i}\left(s_{i}|\theta_{i}^{\mu }\right)$ of an actor online network, the action value $Q_{i}^{\mu^{\prime }}\left(x,a_{1},a_{2},\ldots ,a_{n}\right)$ of a critic target network, and that of agent *i*. Through limited iterations $N_{a}$, the state is combined with stochastic Gaussian noise under the minimum *Q* value that could be excavated. Algorithm 1, to control the sequence, introduces the pseudocode for an agent’s adversarial attack trick.

**Algorithm 1** Adversarial attack trick (with the MADDPG).1: $\mathrm{StateAttack}\left(Q_{i}^{\mu ’}\left(x,a_{1},a_{2},\ldots ,a_{n}\right),\mu_{i}\left(s_{i}|\theta_{i}^{\mu }\right),s_{i}\right)$2: $a_{i}=\mu_{i}\left(s_{i}|\theta_{i}^{\mu }\right),Q_{i}=Q_{i}^{\mu ’}\left(x,a_{1},a_{2},\ldots ,a_{n}\right)$3: **for**
$i=1,N_{a}$**do**4:   $s_{i(\mathrm{noise})}=s_{i}+\mathrm{Gaussian}(0,\sigma_{s}^{2})$5:  $a_{i(\mathrm{noise})}=\mu_{i}(s_{i(\mathrm{noise})}|\theta_{i}^{\mu })$6:  $Q_{i(\mathrm{noise})}=Q_{i}^{\mu ’}\left(x,a_{1},a_{2},\ldots ,a_{i(\mathrm{noise})},\ldots ,a_{n}\right)$7:  **if**
$Q_{i(\mathrm{noise})}<Q_{i}$**then**8:  $Q_{i}=Q_{i(\mathrm{noise})},s_{i}=s_{i(\mathrm{noise})}$9: **end if**10: **end for**11: **return**
$s_{i(\mathrm{noise})}$

The action of agent *i* can be remodeled based on its state after including stochastic Gaussian noise. Similarly, the robustness of the multi-agent intelligent control model could be optimized according to the adversarial attack of all agents by modeling the indeterminacies of the real world.

#### 3.2.2. Adversarial Learning for Cooperators

In addition to uncertain influences from a real environment, an agent is susceptible to strategy changes made by other agents in the overall system [37]. In other words, an agent cannot produce the optimal action strategy to match other agents when those agents change strategies. Our algorithm preprocesses the actions of cooperators using adversarial training techniques, so the agent’s strategy is updated based on the worst decisions of other agents. Specifically, as the neural network is updated, the cumulative return of agent *i* is optimized under the condition that all cooperators use adversarial strategies. The cumulative return of agent *i* combined with adversarial learning can be formulated as:(25)$$\begin{array}{c} E_{x∼\rho^{\mu }}\left(R_{i}\right)=\underset{a_{j\neq i}^{t}}{min}E_{x∼\rho^{\mu }}\left[\sum_{t=0}^{T}\gamma^{t}r_{i}\left(x^{t},a_{1}^{t},\ldots ,a_{n}^{t}\right)\left|a_{i}^{t}=\mu \left(s_{i}^{t}\right)\right.\right]\\ =E_{x^{0}∼\rho^{\mu }}\left[\underset{a_{j\neq i}^{0}}{min}Q_{i}^{\mu }\left(x^{t},a_{1}^{t},\ldots ,a_{n}^{0}\right)\left|a_{i}^{0}=\mu \left(s_{i}^{0}\right)\right.\right] \end{array}$$
where $\rho^{\mu }$ represents the state distribution. That means $x^{t+1}$ would be influenced by the actions of all agents. Furthermore, the action value *Q* function could be defined in a recursive form as:(26)$$Q_{i}^{\mu }\left(x,a_{1},\ldots ,a_{n}\right)=r_{i}\left(x,a_{1},\ldots ,a_{n}\right)+\gamma E_{s^{\prime }}\left[\min_{a_{j\neq i}^{\prime }}\;Q_{i}^{\mu }\left(x^{\prime },a_{1}^{\prime },\ldots ,a_{n}^{\prime }\right)\left|{}_{a_{i}^{\prime }=\mu \left(x_{i}^{\prime }\right)}\right.\right]$$

The single-step gradient descent method was introduced to overcome high computing costs [38]. Using this method, the actions taken by cooperators are those exhibiting a mixed disturbance, and the direction of the disturbance is the orientation in which the *Q* function is decreasing. To summarize, the update process of the critic online network can be formulated as:(27)$$L\left(\theta_{i}\right)=E_{x,a,r,x^{\prime }}\left[{\left(y_{i}-Q_{i}^{\mu }\left(x,a_{1},\ldots a_{n};\theta_{i}^{Q}\right)\right)}^{2}\right]$$
(28)$$y_{i}=r_{i}+\gamma Q_{i}^{\mu^{\prime }}\left(x^{\prime },a_{1}^{\prime *},\ldots ,a_{i}^{\prime *},\ldots ,a_{n}^{\prime *};\theta^{Q^{\prime }}\right)$$
(29)$$a_{i}^{\prime }=\mu_{i}^{\prime }\left(s_{i}\right)$$
(30)$${a^{\prime }}_{j\neq i}^{*}=a_{j}^{\prime }+\eta_{j}$$
(31)$$\eta_{j\neq i}=\arg\min_{\varepsilon_{j\neq i}}\;Q_{i}^{\mu^{\prime }}\left(x^{\prime },a_{1}^{\prime }+\eta_{1},\ldots ,a_{i}^{\prime }+\eta_{i},\ldots ,a_{n}^{\prime }+\eta_{n}\right)$$
where $\theta^{Q^{\prime }}$ represents the critic target network, ${a^{\prime }}_{j\neq i}^{*}$ represents the action of other agents in their minimum conditions, and $\eta_{j\neq i}$ represents the disturbance added for agent *j*. By linearizing the *Q* function, the parameter $\eta_{j}$ is used to denote the gradient direction of $Q_{i}^{\mu }\left(x,a_{1},\ldots ,a_{n};\theta_{i}^{Q}\right)$ at $a_{j}^{\prime }$. $\eta_{j}$ can be replaced with this gradient approximation:(32)$${\tilde{\eta }}_{j\neq i}=-\varepsilon \nabla_{a_{j}}Q_{i}^{\mu^{\prime }}\left(x^{\prime },a_{1}^{\prime },\ldots ,a_{n}^{\prime }\right)$$

The critic network structure of the MADDPG combined with adversarial learning is illustrated in Figure 7. When the MADDPG is implemented, adversarial interference must be added to the actions of other agents without requiring the critic network to be remodeled.

To summarize, the A2-MADDPG algorithm proposed in this paper employs an adversarial attack trick and adversarial learning to process an agent’s state information and other agents’ actions during training. This bridges the gap between simulated training and the real world by adding adversarial disturbances. The overall A2-MADDPG algorithm is described in Algorithm 2.

**Algorithm 2** Adversarial attack trick and adversarial learning MADDPG (A2-MADDPG) algorithm.1: for *N* agents, randomly initialize their critic network $Q_{i}\left(s_{i},a_{i};\theta_{i}^{Q}\right)$ and actor network $\mu_{i}\left(s_{i}|\theta_{i}^{\mu }\right)$2: synchronize target networks $Q_{i}’\left(s_{i},a_{i};\theta_{i}^{Q’}\right)$ and $\mu_{i}’\left(s_{i}|\theta_{i}^{\mu ’}\right)$ with $\theta_{i}^{Q’}\leftarrow \theta_{i}^{Q},\theta_{i}^{\mu ’}\leftarrow \theta_{i}^{\mu }$3: initialize hyperparameters: experience buffer D, mini-batch size *m*, max episode *M*, max step *T*, actor learning rate *l_a_*, critic learning rate *l_c_*, discount factor γ, soft update rate τ4: **for**
*episode* = 1, *M*
**do**5:  reset environment, and receive the initial state x6:  initialize exploration noise of action $N_{\mathrm{action}}$7:  **for**
*t* = 1, *T*
**do**8:  $s_{i(\mathrm{noise})}\leftarrow \mathrm{StateAttack}\left(Q_{i}^{\mu ’}\left(x,a_{1},a_{2},\ldots ,a_{n}\right),\mu_{i}\left(s_{i}|\theta_{i}^{\mu }\right),s_{i}\right)$9:   for each agent *i*, select action $a_{i}=\mu \left(s_{i(\mathrm{noise})};\theta_{i}^{\mu }\right)+N_{\mathrm{action}}$10:   execute $a_{1},a_{2},\ldots ,a_{n}$, rewards $r_{1},\ldots ,r_{n}$, and next state x’11:   store sample $\left(x,a_{1},a_{2},\ldots ,a_{n},r_{1},\ldots ,r_{n},x’\right)$ in D12:   **for**
*agenti* = 1, *n*
**do**13:    randomly extract *m* samples $\left(x^{k},a^{k},r^{k},{x’}^{k},\right)$
14:    update critic network by Equation (Equation 27)15:    update actor network by:
$$\nabla_{\theta_{i}^{\mu }}J=m^{-1}\sum_{j}\left[\nabla_{\theta_{i}^{\mu }}\mu_{i}\left(a_{i}\left|s_{i}\right.\right)\nabla_{a_{i}}Q_{i}^{\mu }\left(x,a_{1}\ldots a_{i}\ldots a_{N};\theta_{i}^{Q}\right)\left|{}_{a_{i}=\mu_{i}\left(s_{i}\right),a_{j\neq i}^{*}=a_{j}^{k}+{\tilde{\eta }}_{j}}\right.\right]$$16:   **end for**17:   update target networks by $\theta_{i}^{Q’}\leftarrow \tau \theta_{i}^{Q}+(1-\tau )\theta_{i}^{Q’},\theta_{i}^{\mu ’}\tau \leftarrow \theta_{i}^{\mu }+(1-\tau )\theta_{i}^{\mu ’}$18:  **end for**19: **end for**

## 4. Experiment and Result Analysis

This section describes the simulation’s settings and a series of experiments implemented to analyze the effectiveness and performance of the approaches proposed in previous sections.

### 4.1. Simulation Environment Settings

The experiments were conducted using Pycharm and the gym module on an Ubuntu16.04 system with an Intel i7-6700K CPU, a GeForce1660Ti graphics card, and 16 G of RAM. As shown in Figure 8, the testbed was a square (20 km on a side) in a two-dimensional plane. Each obstacle was assumed to have a round threat area with radius $r_{ob}\in \left[0.6,1.3\right]$ km (black circle). The attack range of the pursuers (red circles with a UAV inside) was set to 1.2 km, which means the mission was considered successful for pursuers when the distance between them and at least one evader (blue UAVs) was within 1.2 km. Table 1 provides the parameters used for the platforms.

In the DRL-based multipursuer framework, a two-layer perceptron model was constructed for the actor and critic networks. Two fully connected 15 × 64 × 64 × 2 neural networks were created for the actor network and its target. Furthermore, two fully connected 17 × 64 × 64 × 1 neural networks were created for the critic network and its target. Each round ends when the pursuers capture an evader, a platform collides with an obstacle, or the simulation reaches the maximum number of steps. After each round, the environment was reset, and the next round began. Network training ended when the experience buffer was full, and an Adam optimizer was used to determine the neural network parameters. The hyperparameters of the network are shown in Table 2.

### 4.2. Experiment on the Performance of the A2-MADDPG

#### 4.2.1. Performance of Pursuers

To examine the performance of trained pursuers, this study used different algorithms to train them to fight evaders that were trained using the constant MADDPG algorithm from Experiment 1. Specifically, we employed the IMDDPG, MADDPG, and A2-MADDPG algorithms in the MAS of pursuers and present comparative data about the average return values for the last 1000 training episodes in Figure 9.

As illustrated in Figure 9a, UAVs trained using all three algorithms required roughly 8000 episodes to converge to a steady value with an average reward. In Figure 9b,c, the pursuers and evaders play against each other, and their respective average rewards do not converge to a stable state until about 16,000 episodes. The A2-MADDPG achieved higher convergence approaching the average reward for pursuers and a lower result for evaders. This means the proposed algorithm produced agents that were highly successful at pursuing while preventing targets from escaping, which resulted in lower rewards. To prove the real performance of our algorithm, the average time of first pursuit and the average success of the pursuers in the last 1000 training episodes are recorded in Figure 10.

Figure 10a shows that the MAS pursuers can complete their pursuit after being trained using the three DRL algorithms. As the training time increases, A2-MADDPG UAV formations can pursue evaders in a shorter time. This means that the A2-MADDPG model has more advanced co-adjutant siege capabilities. Figure 10b shows the variation of eligible pursuing times produced during training. The pursuers quickly completed a large number of pursuits in each round, but their success gradually decreased as two maneuvering objects were introduced, confronted one another, and stabilized. This means that MADDGP evaders could also make intelligent decisions autonomously to flee. Ultimately, the average eligible pursuing time of A2-MADDPG UAVs was about four per round, which is better than the other two algorithms.

To present the pros and cons of each algorithm in the steady state, the last 10,000 rounds (from 40,000 to 50,000 episodes in Figure 9 and Figure 10) were analyzed. The results are presented in Table 3 and include the average return values for pursuers and evaders, the average and maximum first pursuit time, and the average and maximum successes.

Table 3 shows the performance of each algorithm after stable convergence. The IMDDPG pursuers that used a distributed critic network had an average return value of 42.15, while the MADDPG pursuers had 72.87. Meanwhile, the A2-MADDPG algorithm improved the pursuers’ performance, causing the average return value to rise to 103.20. Driven by the shaped reward function, A2-MADDPG pursuers developed efficient strategies, thereby obtaining a higher round reward. The earliest pursuit time indicator reflects the performance of time-efficient decisions that pursuers made. Table 3 shows that for the A2-MADDPG, the earliest pursuit time became shorter, and its average value was reduced from IMDDPG’s, 32.17 to 24.61. The maximum earliest pursuit time was reduced from 35.662 to 26.802. The pursuer success number describes how many evaders were successful in each round. This indicator was better for the A2-MADDPG than for the other two algorithms, which indicates that the MAS pursuers based on it had better performance.

#### 4.2.2. Performance of Evaders

In Experiment 2, we trained evaders using the IMDDPG, MADDPG, and A2-MADDPG to challenge MADDPG pursuers. The average reward results for the last 1000 training episodes are presented in Figure 11.

As illustrated in Figure 11c, the evaders achieved the highest round average using the A2-MADDPG, followed by MADDPG, and finally, IMDDPG. This means that A2-MADDPG evaders had a larger advantage during confrontations, which helped them avoid being attacked by pursuers more often.

We recorded the earliest completion time and the eligible tracking time of the pursuers in each round to verify the algorithm’s performance, as shown in Figure 12. In Figure 12a, observe that the blue curve has the highest values when the experiment stabilized, that is the first time of pursuing task completion during confrontations between MADDPG pursuers and A2-MADDPG evaders was the longest. This means that A2-MADDPG evaders made efficient decisions when avoiding predators. Figure 12b shows the value of the eligible pursuing time for 50,000 training episodes, from which we observed that the average eligible pursuing time under the confrontation between MADDPG pursuers and A2-MADDPG evaders was the smallest. A2-MADDPG evaders demonstrated better escape strategies. In addition, experimental data from the last 10,000 rounds were analyzed, as presented in Table 4, to control the sequence. The analysis included the average return value of the pursuing UAV formation and evader, the average value of the first time of pursuing task completion, the minimum value of the first time of pursuing task completion, the average eligible pursuing time, and the maximum eligible pursuing time.

Table 4 gives the parameters of each algorithm after stable convergence. Compared with the IMDDPG, which uses a distributed critic network, centralized MADDPG evaders had higher average return values, longer times being pursued, and lower occurrences of being caught. A2-MADDPG evaders, with superior maneuverability, could generate effective actions to flee from capture by pursuers. That means the proposed A2-MADDPG also optimized the evaders’ strategies.

### 4.3. Experiment on the Effectiveness of the A2-MADDPG

#### 4.3.1. Effectiveness of Pursuing

In this section, the algorithm for a specific 3V1 pursuit and evasion confrontation was simulated and analyzed. Table 5 provides the initial positions of the UAVs and obstacles in Experiment 3.

Figure 13 presents the confrontation process of the mission starting from the same initial state. The IMDDPG pursuer formation successfully hit the evader after 39 steps. The MADDPG pursuer formation generated better encirclement strategies, and it took more time to hunt down the target (27 steps). The A2-MADDPG formation adjusted the direction and speed of each UAV to more effectively reach the more maneuverable evader. It took 13 steps for the pursuers to reach the escape target.

The algorithms were examined in a random test environment with stochastic initial states and environments in Experiment 4. Based on the chase game trained in Section 4.2.1, the results regarding task completion for 1000 test episodes are shown in Table 6. The improved A2-MADDPG had a higher mission success rate of 88.9% for pursuers, which was higher than the success rate of the MADDPG’s 75.3% and the IMDDPG’s 70.4%. Moreover, the A2-MADDPG pursuers were able to catch the evader in less time. Compared with the MADDPG and IMDDPG, the average value of first time of pursuing completion was 23.698 for the A2-MADDPG, which shows that the A2-MADDPG pursuers can complete pursuits in less time and the effectiveness of their pursuits was enhanced.

#### 4.3.2. Effectiveness of Escaping

To test the effectiveness of escaping, 1000 random environments were generated in Experiment 5 to compare with the IMDDPG, MADDPG, and A2-MADDPG. The results are presented in Table 7. IMDDPG evaders had a success rate of 17.9%, and A2-MADDPG evaders created better evasion strategies, which resulted in an increased success rate of 31.4%. The maximum value of 30.692 for the average value of the first time of being pursued also proved the effectiveness of the A2-MADDPG for evaders. This means that A2-MADDPG evaders specified more effective escape strategies.

In summary, compared with the IMDDPG and MADDPG, the evaluation indicators of the A2-MADDPG were significantly better under the same hyperparameter and training environment settings; in the same test environment, the pursuit and escape strategies trained by the A2-MADDPG were obviously more robust and more efficient than those trained by the other two algorithms. Therefore, the A2-MADDPG had a superior performance in the experiments.

## 5. Conclusions

In this paper, deep reinforcement learning was applied to multi-agent pursuit–evasion decision-making without building a complicated control system, as is commonly performed in traditional approaches. An elaborate MADDPG-based framework was constructed for providing online decision-making schemes and determining the co-adjutant control of multi-agent systems. By introducing adversarial disturbances, an improved A2-MADDPG was proposed that effectively reduced the influence of errors between models and real scenarios. Introducing an adversarial attack trick optimized the robustness of the multi-agent intelligent control model by incorporating adversarial attacks from all agents. An adversarial learning technique was incorporated into our algorithm to overcome the vulnerability of responding to the changes introduced by other agents. This was performed by processing data in the input layer of a critic network. Experimental results showed that the proposed algorithm improved the performance of both types of players in pursuit–evasion games and that the trained agents could devise effective strategies autonomously in confrontational missions.

We intend to expand the pursuit–evasion missions by changing the number of pursuers and evaders in the future and increasing the number of obstacles to make the environment more complex, so as to evaluate the performance, efficiency, and robustness of our algorithms in a more realistic and dynamic space. In addition, we would like to apply the trained robust strategies to drones or unmanned vehicles, so that they can make decisions based on the environmental information obtained by the cameras with an authentic range. This will accelerate the conversion of this work from virtual digital simulations to real multi-agent systems.

## Figures and Tables

**Figure 1 entropy-23-01433-f001:**
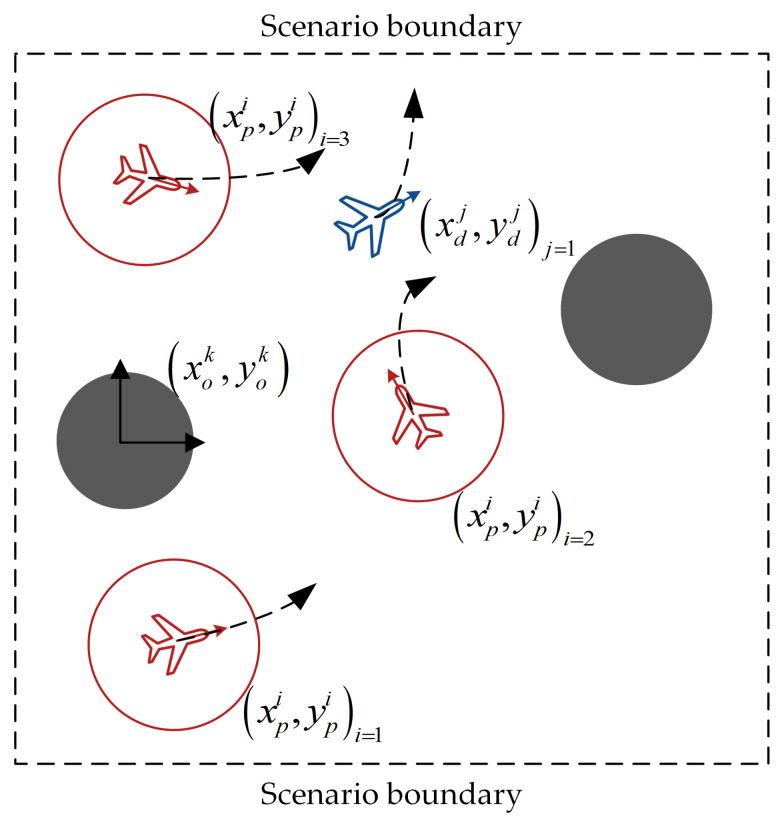
The scenario of multi-agent pursuit–evasion games.

**Figure 2 entropy-23-01433-f002:**
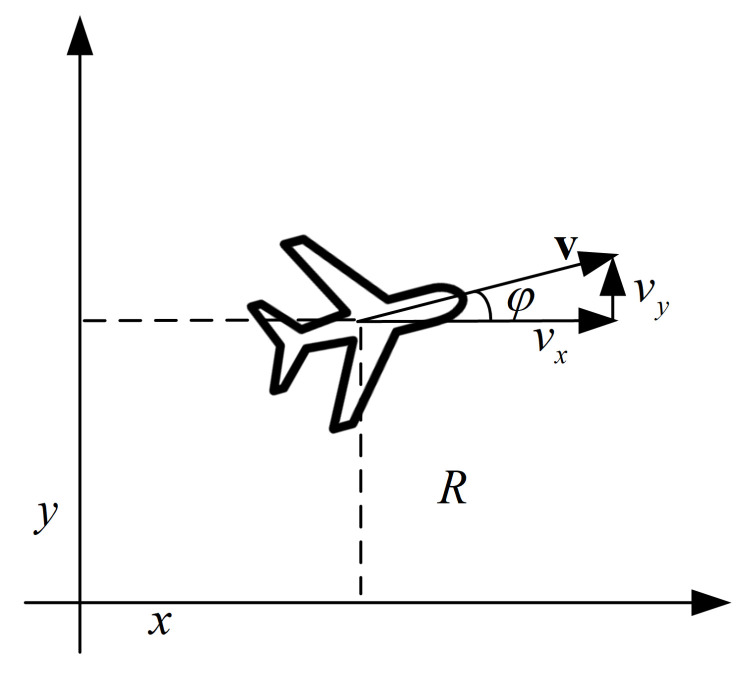
Motion analysis of the UAV.

**Figure 3 entropy-23-01433-f003:**
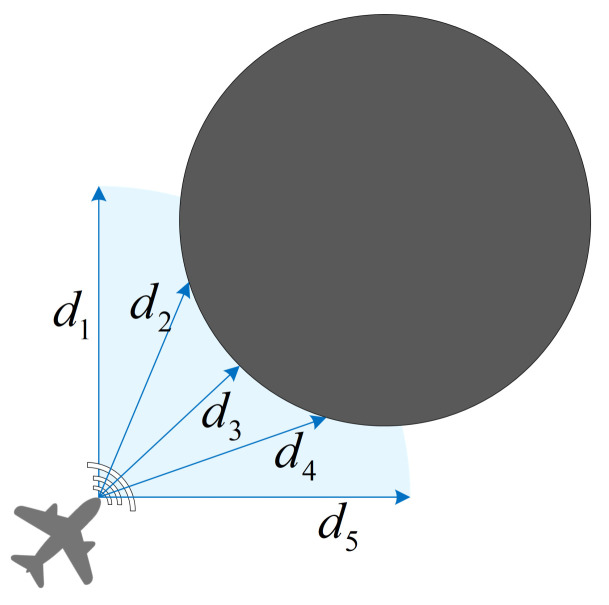
Unmanned system obstacle threat detection based on range sensors.

**Figure 4 entropy-23-01433-f004:**
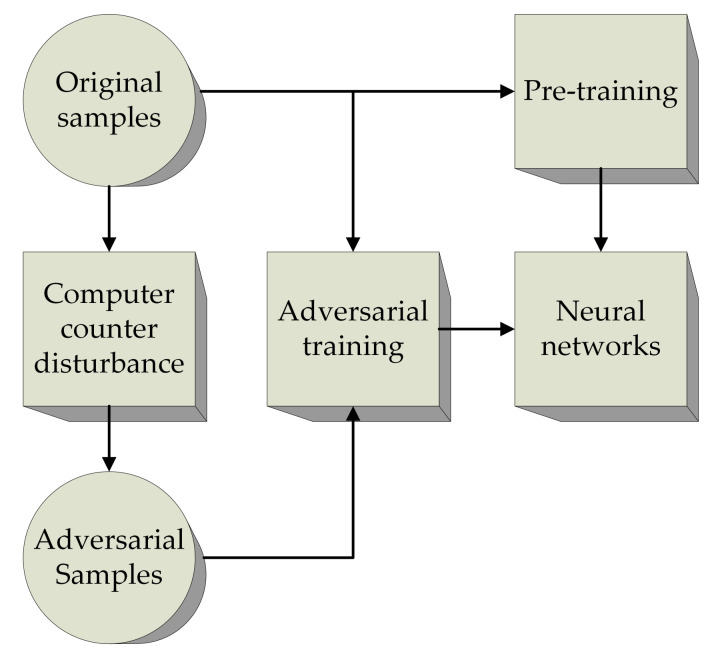
Schematic of adversarial learning.

**Figure 5 entropy-23-01433-f005:**
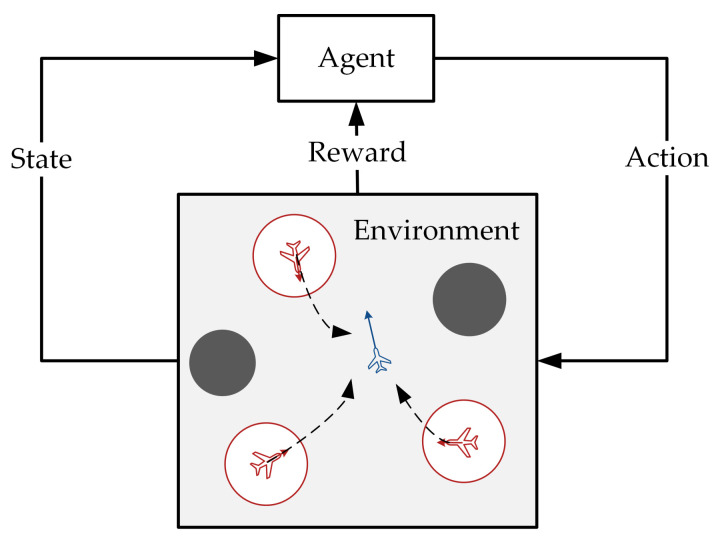
The basic process of deep reinforcement learning.

**Figure 6 entropy-23-01433-f006:**
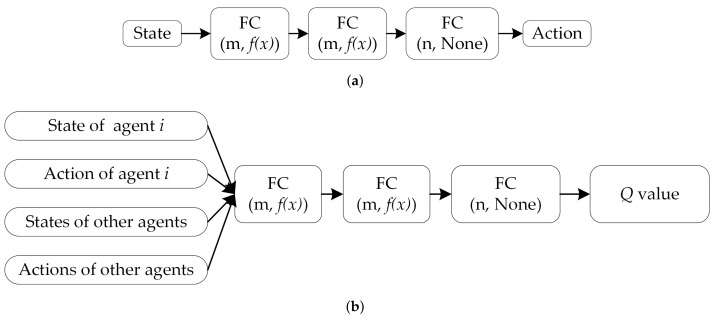
Critic and actor network structures of the MADDPG algorithm. (**a**) Actor and (**b**) critic network structures.

**Figure 7 entropy-23-01433-f007:**
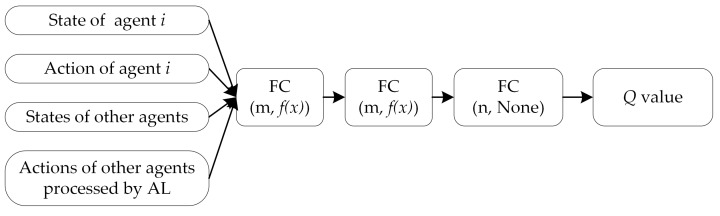
Critic network structures of the MADDPG algorithm combined with AL (the information about the actions of other agents in the input layer is processed by AL).

**Figure 8 entropy-23-01433-f008:**
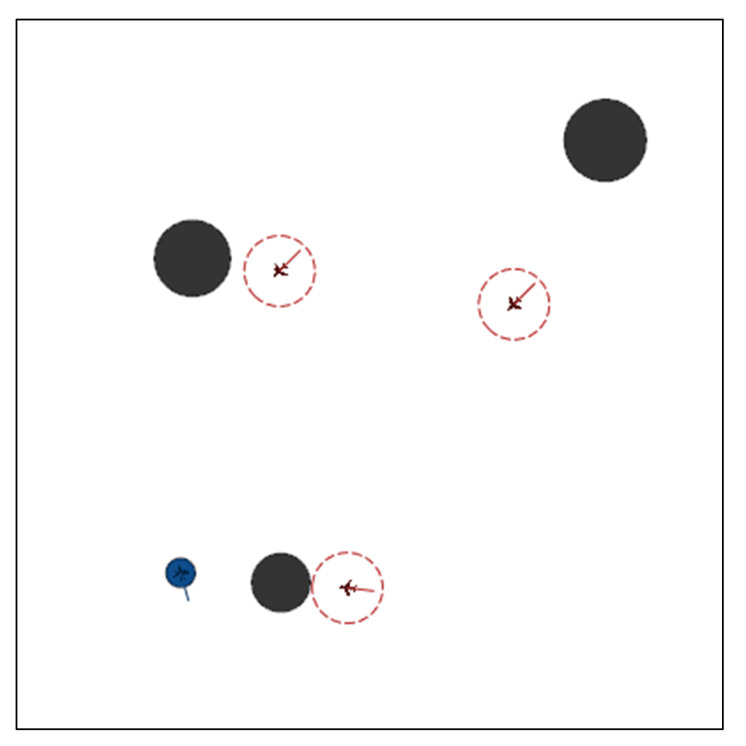
Critic network structures of the MADDPG algorithm combined with AL (the information about actions of other agents in the input layer is processed by AL).

**Figure 9 entropy-23-01433-f009:**
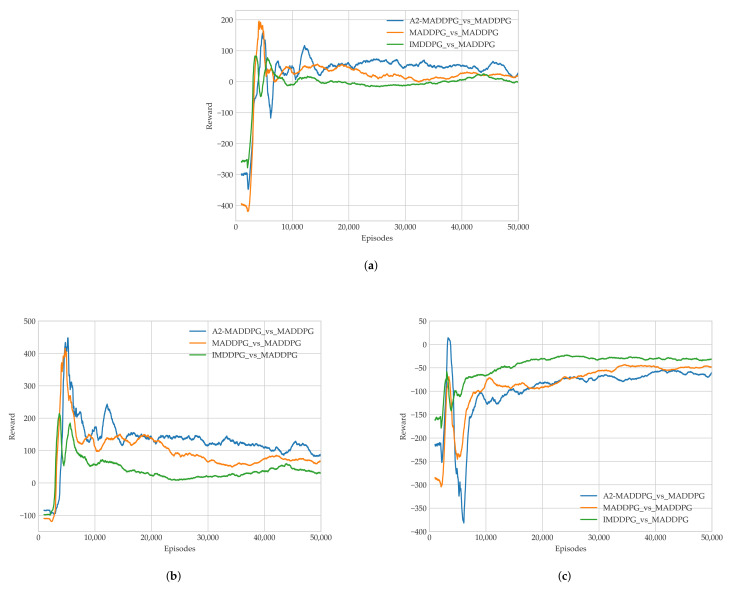
Average rewards in each episode during Training Experiment 1 of (**a**) all agents, (**b**) pursuers, and (**c**) evaders.

**Figure 10 entropy-23-01433-f010:**
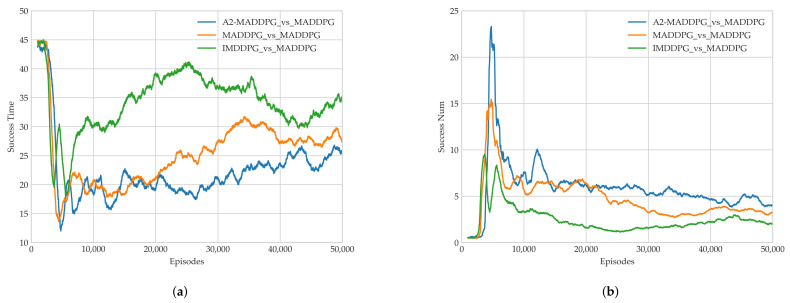
Algorithm performance in Experiment 1. (**a**) Earliest pursuit time and (**b**) successful number of pursuers.

**Figure 11 entropy-23-01433-f011:**
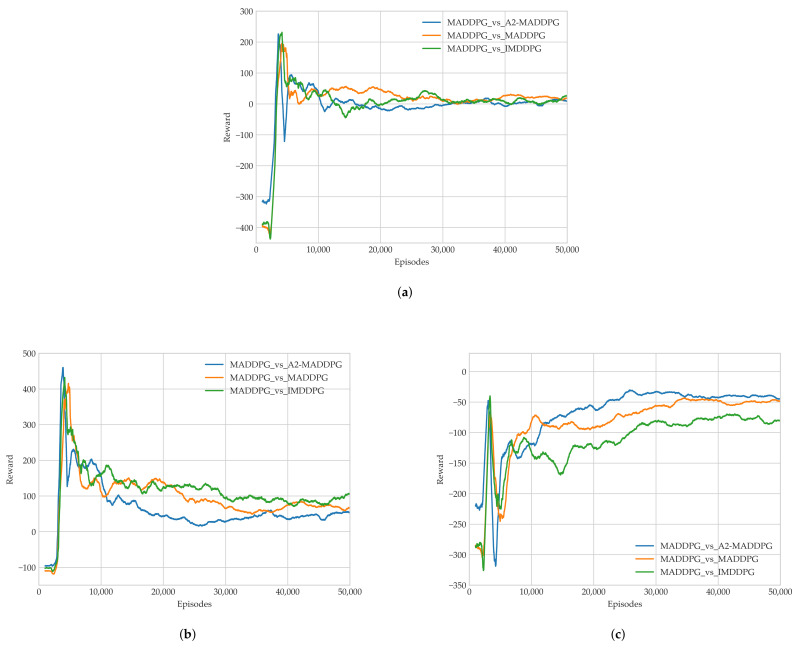
Average rewards in each episode during Training Experiment 2 of (**a**) all agents, (**b**) pursuers, and (**c**) evaders.

**Figure 12 entropy-23-01433-f012:**
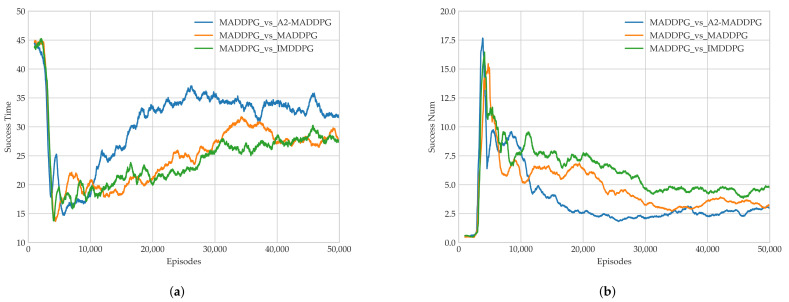
Algorithm performance in Experiment 2. (**a**) Earliest pursuit time and (**b**) number of successful pursuers.

**Figure 13 entropy-23-01433-f013:**
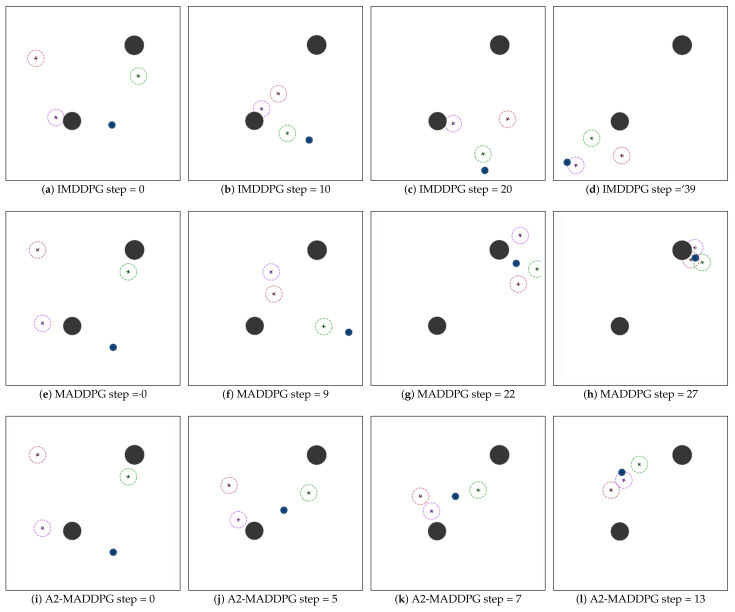
Experiment 3: pursuit–evasion game with one MADDPG-based evader and three groups of pursuers driven by different algorithms. The IMDDPG took 39 steps, MADDPG 27 steps, and A2-MADDPG 13 steps to reach the escape target for the first time.

**Table 1 entropy-23-01433-t001:** The detailed parameter settings of agent platforms in the pursuit–evasion game.

Agent Attributes	Pursuer	Evader
initial velocity	0 km/s	0 km/s
maximum velocity	1 km/s	1.3 km/s
acceleration	0.3 km/s${}^{2}$	0.4 km/s${}^{2}$
detect range	1.5 km	1.5 km
attack range	1.2 km	/

**Table 2 entropy-23-01433-t002:** The detailed parameter setting of agent platforms in the pursuit–evasion game.

Parameters	Values
Experience replay buffer D	100,000
Mini-batch size *m*	1024
Max episode *M*	50,000
Max step *T*	50
Actor learning rate $l_{a}$	0.01
Critic learning rate $l_{c}$	0.01
Discount factor γ	0.95
Soft update rate τ	0.01

**Table 3 entropy-23-01433-t003:** Comparison of the training results (sampled from 40,000 to 50,000 episodes in Experiment 1).

Comparison Standard	IMDDPG	MADDPG	A2-MADDPG
Average return value of pursuers	42.15	72.87	103.20
Average return value of evader	−31.65	−50.47	−60.79
Average value of earliest pursuit time	32.17	27.71	24.61
Maximum value of earliest pursuit time	35.662	29.799	26.802
Average successful number of the pursuers	2.42	3.51	4.46
Maximum successful number of the pursuers	2.994	2.508	5.241

**Table 4 entropy-23-01433-t004:** Comparison of training results (sampled from 40,000 to 50,000 episodes in Experiment 2).

Comparison Standard	IMDDPG	MADDPG	A2-MADDPG
Average return value of pursuers	85.97	72.87	45.13
Average return value of evader	−76.93	−50.47	−40.53
Average value of earliest pursuit time	27.98	27.70	33.07
Maximum value of earliest pursuit time	30.250	29.799	35.819
Average success number of the pursuers	4.45	3.51	2.66
Maximum success number of the pursuers	4.898	3.928	3.115

**Table 5 entropy-23-01433-t005:** Initial positions of UAVs and obstacles in Experiment 3.

Elements	Position
Pursuer 1 (red UAV)	(−7.3, 7.7)
Pursuer 2 (green UAV)	(4.2, 4.0)
Pursuer 3 (purple UAV)	(−8.0, −2.9)
Evader (blue UAV)	(2.0, 8.0)
Obstacle 1 (black circle)	(6.0, 7.0)
Obstacle 2 (black circle)	(3.0, 4.0)

**Table 6 entropy-23-01433-t006:** Effectiveness of pursuing in Experiment 4.

Comparison Standard	A2-MADDPG	MADDPG	IMDDPG
Critic framework of pursuers	Centralized based on AL	Centralized	Distributed
Critic framework of evader	Centralized	Centralized	Centralized
Success rate of pursuers (%)	88.9	75.2	70.4
Average value of the earliest time of pursuing completion	23.698	28.181	32.219

**Table 7 entropy-23-01433-t007:** Effectiveness of escaping in Experiment 5.

Comparison Standard	A2-MADDPG	MADDPG	IMDDPG
Critic framework of pursuers	Centralized	Centralized	Centralized
Critic framework of evader	Centralized based on AL	Centralized	Distributed
Success rate of pursuers (%)	31.4	24.7	17.9
Average value of the earliest time of being pursued	30.692	28.181	32.219

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
