# Peer review of "An Improved Approach towards Multi-Agent Pursuit–Evasion Game Decision-Making Using Deep Reinforcement Learning"

_entropy, 2021, doi:10.3390/e23111433_

Round 1
Reviewer 1 Report
The paper presents an interesting comparison between different algorithms applied in an environment sensing and decision-making in pursuit-evasion games with multi-agent cooperative decision-making.
The approach uses deep reinforcement learning features: MADDPG, adversarial attack trick for agents and optimised technique based on adversarial learning.
-----------------------------------------------
# Key Issues:
1. What is main the novelty of the approach presented in the paper? Is it the use of adversarial disturbances? I would suggest making it clear for the reader by adding the main contributions at the end of the Introduction.
2. Algorithm 2 (page 12).
> The description of Algorithm 2 on page 12 is far from an actual algorithm representation.
There is a lot of text in the algorithm, which seems much more the comments of a code, than an actual code for an algorithm. For example, see the "if" condition in line 12.
-----------------------------------------------
As it follows, some specific comments and corrections:
# 1. Introduction
> Lines 89-90, the authors say the proposed methods have superior performance. Ok, but compared to what?
> I suggest that at the end of the Introduction the major contributions of the paper are clearly described.
# Section 2:
> At the beginning of subsection 2.1.2, it is mentioned the number of pursuers, evaders, and obstacles in the environment. These obstacles can only bring trouble for the evaders? Or also for the pursuers?
# Section 3:
> line 176 mentions "a game that contains an uncertain environment". What do you mean by uncertain? Because Fig. 1 seems to represent an environment that has even a boundary.
> page 9, in the second line (lines are not numbered!) there is a typo in "trust".
> The description of Algorithm 2 on page 12 is far from an actual algorithm representation. There is a lot of text in the algorithm, which seems much more lines of comments of a code, than an actual code for an algorithm. For example, see the if condition in line 12.
Reviewer 2 Report
Authors propose an online decision technique based on deep reinforcement learning to address the problem of environment sensing and decision making in pursuit-evasion games. A control-oriented framework developed from the DRL-based multi-agent deep deterministic policy gradient (MADDPG) algorithm was built to implement multi-agent cooperative decision-making to overcome the limitation of tedious state variables required for the traditionally complicated modeling process. It also proposed a novel adversarial attack trick and adversarial learning MADDPG (A2-MADDPG) algorithm. By introducing an adversarial attack trick for the agents, themselves, uncertainties of the real world are modeled, thereby optimizing robust training.
The idea presented in the paper has a good potential for being appreciated and cited, but it requires some improvements and also extension.
Comments:
- Highlight in what measure and in what parameters, the proposed methodology was found better as compared to existing ones, in the Introduction.
- Introduction should specify more information with regard to the problem definition and scope of the paper.
- A subsection/paragraph related to the problem analyzed should be included. The connection between the problem and the solution proposed is also not pointed out.
- Further papers should be added to the section Literature Review. And each paper should clearly specify what is the proposed methodology, novelty and results cum experimentation. At the end of related works, highlight in some lines what overall technical gaps are observed in existing works, that led to the design of the proposed approach. Considering the scope of the paper, the following paper should be considered:
- http://citeseerx.ist.psu.edu/viewdoc/download?doi=10.1.1.83.102&rep=rep1&type=pdf
- https://books.google.it/books?hl=en&lr=&id=285RhxOtpiUC&oi=fnd&pg=PA220&dq=info:F-z7jb-koH0J:scholar.google.com&ots=5EcF1w_0N1&sig=5hK-RjRi6hVx4Wm032t8dbxS9XY&redir_esc=y#v=onepage&q&f=false
- https://ieeexplore.ieee.org/abstract/document/5493456
- Add Objectives of the paper in points before organization.
- Future scope of the approach should be highlighted and specified better.
Round 2
Reviewer 2 Report
The authors addressed all my comments, then I thinks that it can be accepted in present form.